# Effects of Creatine Supplementation on Histopathological and Biochemical Parameters in the Kidney and Pancreas of Streptozotocin-Induced Diabetic Rats

**DOI:** 10.3390/nu14030431

**Published:** 2022-01-19

**Authors:** Meline Gomes Gonçalves, Matheus Anselmo Medeiros, Licyanne Ingrid Carvalho de Lemos, Lucia de Fátima Campos Pedrosa, Pedro Paulo de Andrade Santos, Bento João Abreu, João Paulo Matos Santos Lima

**Affiliations:** 1Biochemistry and Molecular Biology Graduate Program, Biosciences Center, Federal University of Rio Grande do Norte, Natal 59078-970, RN, Brazil; gomesmeline@gmail.com; 2Bioinformatics Graduate Program, Digital Metropolis Institute, Federal University of Rio Grande do Norte, Natal 59078-400, RN, Brazil; anselmomedeiros50@gmail.com; 3Graduate Program in Nutrition, Federal University of Rio Grande do Norte, Natal 59.078-970, RN, Brazil; licyannelemos@outlook.com (L.I.C.d.L.); lfcpedrosa@gmail.com (L.d.F.C.P.); 4Structural and Functional Biology Graduate Program, Biosciences Center, Federal University of Rio Grande do Norte, Natal 59078-970, RN, Brazil; ppdasantos@gmail.com; 5Biosciences Center, Morphology Department, Federal University of Rio Grande do Norte, Natal 59078-970, RN, Brazil; abreubj@gmail.com

**Keywords:** diabetes mellitus, streptozotocin, creatine, histopathological and histomorphometric analyses, antioxidant response

## Abstract

Diabetes mellitus (DM) is a worldwide health concern, and projections state that cases will reach 578 million by 2030. Adjuvant therapies that can help the standard treatment and mitigate DM effects are necessary, especially those using nutritional supplements to improve glycemic control. Previous studies suggest creatine supplementation as a possible adjuvant therapy for DM, but they lack the evaluation of potential morphological parameters alterations and tissue injury caused by this compound. The present study aimed to elucidate clinical, histomorphometric, and histopathological consequences and the cellular oxidative alterations of creatine supplementation in streptozotocin (STZ)-induced type 1 DM rats. We could estimate whether the findings are due to DM or the supplementation from a factorial experimental design. Although creatine supplementation attenuated some biochemical parameters, the morphological analyses of pancreatic and renal tissues made clear that the supplementation did not improve the STZ-induced DM1 injuries. Moreover, creatine-supplemented non-diabetic animals were diagnosed with pancreatitis and showed renal tubular necrosis. Therefore, even in the absence of clinical symptoms and unaltered biochemical parameters, creatine supplementation as adjuvant therapy for DM should be carefully evaluated.

## 1. Introduction

Diabetes mellitus (DM) is a chronic metabolic disease and a worldwide health concern. In 2019, the estimated number of cases was 463 million, and recent reports project that this number will reach 578 million by 2030, or even 700 million by 2045. A hyperglycemic state characterizes DM, usually due to a reduction in circulating insulin levels or a deficit in the tissue effects of this hormone, or both [1,2]. This reduction occurs when the pancreas does not produce insulin or when the body cannot use it efficiently [3,4]. The most prevalent types are type 1 and type 2 diabetes mellitus (T1DM and T2DM, respectively). T1DM accounts for 5–10% of cases, and Brazil ranks third in the world for new and existing patients of this type [2]. It is characterized by impaired insulin production due to the partial or total destruction of the beta (β) cells, resulting in a progressive deficit of insulin production.

An autoimmune process mediates the destruction of the pancreatic β-cells. However, there are cases in which there is no evidence of autoimmunity, such as idiopathic and minority forms of T1DM. Thus, the patients will produce little or no insulin and, consequently, require exogenous administration of this hormone [1,2,5]. T2DM is the most common type and affects at least 90% of all diabetes cases. It is a complex polygenic disease with genetic inheritance and a significant contribution from environmental factors. Dietary habits and physical inactivity contribute to obesity and are the main risk factors [6]. In this type, there is a relative deficiency of insulin or a resistance to its biological action. Depending on the severity, physical activity and dietary planning can impair the disease progression [1,2,5].

DM induces alterations in the metabolism of carbohydrates, lipids, and proteins, as well as structural changes that lead to several vascular effects, including microangiopathies (retinopathy, nephropathy, and neuropathy) and macroangiopathies such as coronary heart disease and peripheral arterial insufficiency [7,8]. The literature also reports that a severe imbalance in redox homeostasis plays an essential role in developing damage to vascular endothelial cells and can affect both the macro and microvascular spectrum [9]. This redox imbalance, allied to hyperglycemia, also induces a significant increase in the formation of advanced glycation end-products (AGEs), which promotes even more oxidative stress to cells and alters structures morphological and functional properties. In turn, it also increases the expression of inflammatory mediators [10,11]. Therefore, they are closely related to micro and macrovascular complications and contribute to pancreatic β-cell dysfunction [12,13,14].

Several studies have evaluated adjuvant therapies that can help the standard treatment and mitigate DM effects [15,16,17,18,19]. Nutritional supplements have received much attention for fulfilling this purpose, due to their effects on metabolism, leading to improved glycemic control [20,21,22,23]. For this reason, creatine supplementation has been suggested as a possible adjuvant therapy for DM [22,23]. Creatine (α-methyl-guanidino acetic acid) is an imino acid naturally produced by animal metabolism, having as precursors the amino acids arginine, glycine, and methionine, or exogenously obtained through the ingestion of meat, fish, and other animal foods, as well as supplementation [24,25,26].

The literature still debates creatine as a supplement, though less frequently [27,28,29,30]. Some studies have reported that the compound could have harmful effects on kidney function [31,32,33]. On the other hand, other researchers had not identified a risk of renal damage, and episodes of nephrotoxicity are scarce in healthy patients supplemented with creatine [34,35,36,37,38] However, there is a notable scarcity in the histopathological evaluation of compromised organs in DM, especially with creatine supplementation. Therefore, we aim to evaluate whether there are changes in renal and pancreatic tissue’s histological characteristics to elucidate the clinical and histopathological aspects of creatine supplementation in streptozotocin (STZ)-induced T1DM rats. We also assess the cellular oxidative alterations from renal tissues and demonstrate any correlation between the biochemical and histological parameters evaluated.

## 2. Materials and Methods

### 2.1. Study Design

The effect of creatine supplementation was evaluated in adult male Wistar rats (*Rattus norvegicus albinus)* (*n* = 32) for 40 days. We divided the animals into groups of three or four, with water and feed offered ad libitum, and under 12 h light/dark cycle conditions, with an average temperature of 24 ± 2 °C. Then, we randomly allocated to four groups, with a factorial experimental design (2 × 2): (C) normoglycemic animals without creatine supplementation; (CCr) normoglycemic animals with creatine supplementation; (D) diabetic animals induced with STZ without creatine supplementation; (DCr) diabetic animals induced with STZ and supplemented with creatine.

After fasting for fourteen hours, the groups D and DCr received a single dose intraperitoneal injection of STZ (40 mg/kg) dissolved in 10 mM sodium citrate buffer, pH 4.5, for diabetes induction [39]. After seven days, we measured fasting blood levels, retrieved from the tail vein with the aid of a portable glucometer (ONETOUCH^®^ ULTRA^®^) to confirm induction. Animals with blood glucose greater than 250 mg/dL were considered diabetic.

The CCr and DCr groups received creatine supplementation through isocaloric solid ration in the following two phases: the saturation phase, five days before DM1 induction, and the maintenance phase, during the thirty-five days of the experiment, with 13% and 2% (g/kg of feed) creatine monohydrate, respectively [28,40,41,42]. Then, based on the daily-measured feed consumption, the percentage of creatine contained in the different phases (saturation and maintenance) for the CCr and DCr groups, and the number of animals in these groups, we calculated the creatine consumption (g) per animal per day and the creatine consumption (g) per Kg of animal. These doses correspond to about 13.5–15 g kg^−1^ and about 2.4–4.2 g kg^−1^, during the saturation and maintenance phases, respectively. We based the used creatine-supplementation levels on the ones reported elsewhere [42,43,44,45], considering that rats have a basal metabolic rate (BMR) about 6.4 times greater than humans [42,46].

On the forty-first day of the experiment, the animals were anesthetized with isoflurane and euthanized by thoracotomy. We retrieved blood samples by hepatic portal vein puncture to analyze the following biochemical parameters: aspartate transaminase (AST), alanine aminotransferase (ALT), fasting glucose, urea, and creatinine. In addition, the pancreas and kidneys were collected and stored in buffered formalin (10% *v*/*v*) for later morphological analysis. We also retrieved kidney tissues samples to assess antioxidant responses and cellular oxidative status parameters. The Ethics Committee on the Use of Animals from the Federal University of Rio Grande do Norte state (CEUA-UFRN, Natal/RN) approved the present study under protocol No. 030.025/2017.

### 2.2. Evaluation of Biochemical Parameters

Capillary blood glucose was checked weekly through the tail vein with a portable glucometer (ONETOUCH^®^ ULTRA^®^). In addition, at the end of the experiment, we analyzed the concentrations of glucose, serum creatinine, and urea, AST, and ALT levels from hepatic portal vein blood samples. After 6 h of storage in test tubes, we centrifuged the blood samples (5000× *g*) for 15 min to collect the serum. Specific colorimetric enzymatic tests were used to quantify each parameter (Labtest, Lagoa Santa, MG, Brazil), following the manufacturer’s protocol, and we measured absorbance values using a spectrophotometer.

### 2.3. Morphological Analysis of Pancreas and Kidney Tissues

The pancreas and kidney stored in buffered formalin (10%) were dehydrated after 24 h in an increasing alcoholic series (70%, 80%, and 90%), diaphanized in xylol, impregnated, and embedded in histological paraffin. Subsequently, using a microtome, we sectioned 5 µm fragments and stretched them on a glass slide, and stained them with routine staining (hematoxylin and eosin). The kidneys samples were also stained with PAS. After covering with a coverslip, we analyzed the slides under a light microscope. The analysis was performed at Instituto Santos Dumont (ISD) using a binocular microscope from the manufacturer Carl Zeiss (Axio Imager.Z2) with a camera attached from the same manufacturer (AxioCam MRc5). We scanned the slides in 6 fields on both the y and *x*-axis, totaling 36 fields for each slide, using the Stereo Investigator^®^ software from MBF Bioscience. Then, to perform tissue histomorphometry, the ImageJ 1.52a program (National Institutes of Health, Bethesda, MD, USA) was used to count and measure the areas, in mm^2^, of the pancreatic islets and renal glomeruli.

The evaluation of tissue damage, through histopathological analyses, was performed blindly by an expert pathologist using a score (0: absent, 1: mild, 2: moderate, 3: intense for each histopathological finding), who observed signs of damage, such as parenchymal atrophy, pancreatic duct alteration, ductal ectasia, fibrosis, acinar destruction, adipose replacement, hyperemia, pancreatic islet hyperplasia, inflammatory infiltrate and necrosis in the pancreas laminae. We evaluated renal injury by the following parameters: signs of thickening of the basement membrane, glomerular ischemia, glomerular and arteriolar hyalinosis, intercapillary glomerulosclerosis, inflammatory infiltrate, proteic, necrotic, atrophic, and hypertrophic tubules.

### 2.4. Protein Extraction and Antioxidant Enzymes Activities

Total soluble proteins from renal tissues were extracted using 50 mM potassium phosphate buffer (pH 7.0) containing 1 mM L-ascorbic acid and 0.1 mM EDTA, following centrifugation at 13,000× *g* (Hettich Centrifuge. Model MI-KRO 200R. Tuttlingen, Germany) for 20 min at 4 °C. The supernatant was filtered and collected as the total soluble protein fraction and stored at −20 °C. We used the method described by BRADFORD (1976) to estimate the concentration of total soluble proteins [47].

We measured catalase (CAT) activity (EC 1.11.1.6) by the hydrogen peroxide absorbance at 240 nm for 5 min [48], using a UV-spectrophotometer (Amersham Biosciences. Model: Ultrospec 2100 pro. Cambridge, UK). The enzyme activity was calculated from the consumption of H_2_O_2_, using the molar extinction coefficient of 40 mM^−1^ cm^−1^, calculated under the same conditions. The activity of superoxide dismutase (SOD) (EC 1.15.1.1) was determined by inhibiting the reduction in NBT (Nitroblue tetrazolium) by the protein extract, thus preventing the formation of the chromophore. We carried out the essay on a 50 mM potassium phosphate buffer (pH 7.8), with 0.1 mM EDTA, 13 mM L-methionine, and 75 μM of NBT. The reaction started after adding riboflavin and transferring the reaction to a chamber illuminated by a fluorescent lamp for 1 min and then read at 560 nM in a spectrophotometer (Femto 700 plus-São Paulo, SP, Brazil). We determined the activity by calculating the amount of extract that inhibited 50% of NBT reduction and expressed it as activity unit (AU) g^−1^ FM min^−1^ [49]. The activity of glutathione peroxidase (GPx) (EC 1.11.1. 9) was determined in potassium phosphate buffer 50 mM, pH 7.0, containing 10 U/mL glutathione reductase (GR), 1 μM reduced glutathione (GSH), 250 nM nicotinamide adenine dinucleotide phosphate (NADPH), and H_2_O_2_ at 2.5%. We measured the reactions at 340 nm every 30 s for 5 min.

### 2.5. Acid Extraction, H_2_O_2_ Content, Lipid Peroxidation, and Thiol Content

Kidney samples were ground in a mortar with liquid nitrogen and transferred to 1% trichloroacetic acid (TCA) solution with 0.5% activated carbon. Each mixture was centrifuged at 13,000× *g* at 4 °C for 25 min (Hettich Centrifuge. Model MIKRO 200R. Tuttlingen, Germany), and the supernatant was collected and stored for further assays [50].

The H_2_O_2_ content of the samples was measured using the TCA extracts through the specific reaction of H_2_O_2_ with potassium iodide (KI) [51]. We performed the assay on 10 mM potassium phosphate buffer, pH 7.0, in the presence of 1 M KI solution added to the extracts and put in the dark for 1 h, and measured the absorbance at 390 nm in a spectrophotometer (Amersham Biosciences. Model: Ultrospec 2100 pro. Cambridge, England) against a reference blank composed of 1% TCA replacing the tissue extract. The H_2_O_2_ content was calculated using a standard reaction curve, and we expressed the results in μmol H_2_O_2_. g^−1^ FM. Lipid peroxidation was measured using the TBARS method, using the molar extinction coefficient of 155 mM^−1^ cm^−1^ for the MDA-TBA complex and the results were expressed as nmol g^−1^ FM [52].

The content of soluble thiols was determined using the colorimetric method of Ellman (1959) and Riddles (1979) [53,54], by the proteic sulfhydryl groups (SH) reaction with 5,5′-dithiobis (2-nitrobenzoic acid) (DTNB), following reads at 412 nm. The total thiol content of each sample was determined from a cysteine standard curve and expressed as μM/mg protein cysteine equivalents.

### 2.6. Determination of Carbonylated Protein Content

We estimated carbonyl derivatives by reacting to 2,4-dinitrophenylhydrazine (DNPH), generating hydrazone, following absorbance reads at 370 nm [55]. The results were expressed in nmol of carbonylated protein per mg of protein. We used the molar extinction coefficient of DNPH (22,000 M^−1^ cm^−1^) to calculate the content of incorporated DNPH.

### 2.7. Statistical Analysis

During the experiment, one animal from each diabetic group (D and DCr) died. Thus, these groups had *n* = 7 each for analysis. All data were expressed as mean ± standard error of the mean (SEM). A normality test was performed for the data using the Kolmogorov–Smirnov test. We used the one-way ANOVA parametric test, followed by Tukey’s post-test for multiple comparisons. We used two-way ANOVA and Tukey’s post-test for repeated measures, such as water and feed consumption. *p* values ≤ 0.05 were considered significant.

## 3. Results

### 3.1. Evaluation of Physiologic and Clinical Parameters

Based on the daily feed consumption, we estimated the average creatine dose per animal in the supplemented groups. The CCr group consumed an average of 3.38 ± 0.13 and 0.56 ± 0.04 g of creatine/animal/day, respectively, in the saturation and maintenance phases. The DCr group consumed an average of 3.33 ± 0.15 and 0.79 ± 0.04 g of creatine/animal/day, respectively, in the saturation and maintenance phases. There was no significant difference between the CCr and DCr groups in creatine consumption in both phases. Regarding water consumption and feed (Figure 1A,B), we observed that D and DCr groups presented polydipsia and polyphagia after diabetes confirmation (*p* < 0.001). In both parameters, there was no significant difference between the supplemented and unsupplemented animals. The evolution of the weight gain percentage (Figure 1C) demonstrated weight gain for groups C and CCr during the experiment, while groups D and DCr lost weight significantly (*p* < 0.001).

As expected, the diabetic groups had significantly increased blood glucose when compared to controls (*p* < 0.001) at the end of the experiment (Figure 2A). However, the DCr group blood glucose levels were significantly lower than those observed for the D group (*p* < 0.05). Although we did not observe significant changes in serum creatinine concentrations among all experimental groups (Figure 2B), there was a significant increase (*p* < 0.001) in serum urea levels in groups D and DCr compared to the non-diabetic ones (Figure 2C). The DCr group had significantly lower (*p* < 0.001) serum urea concentrations than the D group, similar to the fasting blood glucose. Serum levels of AST (Figure 2D) and ALT (Figure 2E) were also analyzed. The former parameter did not vary significantly between groups. Diabetic groups presented significant increases (*p* < 0.001) in ALT levels, though the creatine-supplemented one had significantly lower ALT concentrations (*p* < 0.01) compared to the D group. All of these results validate the experimental model and treatment protocol used.

### 3.2. Morphological Analyses of the Pancreatic Tissue

We divided the morphological analyses into histomorphometry and histopathological parameters. Regarding histomorphometry, the creatine-supplemented diabetic animals (DCr group) did not present significant differences in the number of pancreatic islets when compared to the C and CCr groups (*p* > 0.05) (Figure 3A). However, both diabetic groups (D and DCr) showed a significant reduction in the area of the pancreatic islets when compared to non-diabetic ones (*p* < 0.001). Even so, creatine supplementation seems to lessen the effects on the islets since the reduction in their area was significantly smaller in the DCr group compared to the D group (Figure 3B).

As for the histopathological analysis, there was no statistical difference (*p* > 0.05) between the groups for parenchymal atrophy (*p* = 0.2468), pancreatic duct changes (*p* = 0.4682), and islet hyperplasia (*p* = 0.1012), although we observed these alterations in some animals (Figure 3C–F).

As for the findings of acinar destruction, ductal ectasia, and fibrosis, there was a significant difference for groups D and DCr in all these parameters compared to group C (Figure 4). Animals in groups D and DCr had a more significant number of areas with destroyed serous acini (*p* < 0.05 and *p* < 0.001, respectively) when compared to C and CCr groups (Figure 4A).

As for ductal ectasia and fibrosis (Figure 4B,C), we observed a similar pattern when comparing the groups, in which the animals in groups C and CCr did not differ from each other. However, the creatine-supplemented control group showed more areas with these histological findings, resulting in a non-significant difference from group D. Still, for ductal ectasia (Figure 4B), the groups D and DCr did not differ statistically, while for fibrosis (Figure 4C), the DCr group had more affected areas, differing statistically from the other experimental groups (*p* < 0.001). It is possible to identify acinar destruction (Figure 4F), ductal ectasia (Figure 4G) and fibrosis (Figure 4H), respectively, in photomicrographs of the histological slides.

The fat replacement and hyperemia (Figure 4D,E, respectively) observations were somewhat interesting. It is possible to identify that the animals in the CCr group presented a statistical difference between the other groups regarding the number of affected areas (*p* < 0.05). The animals in the DCr group also showed a higher number, but without significant difference between groups C and D. We also evidenced adipose replacement in the parenchyma from the pancreatic tissue (Figure 4I).

We also assessed inflammatory cell infiltrates in the pancreatic tissue (Figure 5C,D). Animals from groups D and DCr (*p* < 0.01) showed an increased number of these infiltrates. Although creatine-supplemented non-diabetic animals (CCr group) also presented inflammatory cell infiltrates, this result was not statistically significant to the non-supplemented controls (C). As expected, the control animals did not present any area with foci of inflammatory infiltrate (Figure 5A). In addition to these findings, there was necrosis of the pancreatic parenchyma in the animals of groups CCr, D, and DCr (Figure 5B,E,F), but unlike the STZ-treated group (D), the animals in groups CCr and DCr did not differ significantly from the controls. We did not observe any necrosis spots in the pancreatic tissues from the control animals (Figure 5B).

### 3.3. Morphological Analyses of Renal Tissues

We used the same division into histopathological and histomorphometric parameters for the analyses of renal tissues. All the histological slides from all groups did not exhibit the presence of arteriolar hyalinosis, atrophic and hypertrophic tubules. Due to differences in water ingestion between the experimental groups, we measured the relative kidney weight (Figure 6C). There was significant kidney hypertrophy (*p* < 0.001) of the D and DCr groups compared to the C and CCr groups. Interestingly, the DCr group showed significantly greater renal hypertrophy (*p* < 0.001) than the D group.

In the renal histomorphometric analysis, regarding the number of glomeruli, there was no statistical difference between the groups, except for the animals in the DCr group, which showed a significant decrease compared to groups C and CCr and group D (*p* < 0.05) (Figure 6A). As for the glomerular area, the animals in group D showed a significant reduction compared to groups C and CCr (*p* < 0.05). Group DCr showed a significant increase compared to group D and showed no statistical difference compared to the control groups (Figure 6B). Unexpectedly, only the animals in the DCr group differed statistically from the others (*p* < 0.01) and presented a higher number of observations of glomerular hyalinosis (Figure 6D). We did not evidence a significant difference in the thickening of the basal membrane between the experimental groups.

The renal glomeruli of the animals in groups CCr, D, and DCr presented ischemia (Figure 7). The CCr group showed a statistical difference (*p* < 0.05) when compared to group C. Groups D and DCr had a higher intensity in ischemic findings compared to group C, but without significantly diverging from CCr (Figure 7A). We also found significant diffuse intercapillary glomerulosclerosis in the diabetic animals’ kidneys (D and DCr) (Figure 8). 

Another histological finding noticed in the renal tissues was cylindrical protein tubules or hyaline cylinders in at least one slide of all experimental groups (Figure 9). Interestingly, only the CCr group had an increased measurement (*p* < 0.01) (Figure 9A). We also observed tubular necrosis in the animals of groups CCr, D, and DCr (Figure 9B,E,F). There was a significant difference between the animals in group CCr and the other groups (*p* < 0.01). Moreover, the diabetic non-supplemented animals (Group D) differed statistically from the C and CCr groups (Figure 9A). As in the pancreatic tissue, renal tissues also presented inflammatory infiltration (Figure 9G), but the difference between the groups was not significant. 

### 3.4. Antioxidant Enzyme Activities and Redox State Parameters from Renal Tissue

At the end of the experiment, we measured the activity of antioxidant enzymes from protein extracts from kidney tissue sections. Non-supplemented diabetic animals (D group) showed a significant decrease in the CAT (Figure 10A), SOD (Figure 10B), and GPX (Figure 10C) activities when compared to other experimental groups. Surprisingly, there was no statistical difference between the control groups and the DCr group in these quantified enzymes. We also measured the hydrogen peroxide content (Figure 10D). There was a significant increase in this parameter (*p* < 0.01) in group D compared to groups C and CCr. This same difference was not observed in the DCr group, which did not show significant differences compared to the control groups and showed significantly lower H_2_O_2_ content (*p* < 0.001) compared to the D group.

When quantifying the content of thiobarbituric acid reactive substances (TBARS) (Figure 10E), the CCr group showed a significant increase (*p* < 0.001) in lipid peroxidation in comparison to group C and both diabetic groups (D and DCr). Again, creatine-supplemented diabetic animals (DCr) did not present significant differences from the control values in the carbonylated protein content (Figure 10F). However, unexpectedly, diabetic animals showed a significant decrease (*p* < 0.05) compared to the other groups in this same oxidative status parameter. Moreover, we did not observe any difference between the soluble thiol content between the experimental groups. 

## 4. Discussion

To the best of our knowledge, this was the first study that evaluated morphological aspects, both histopathological and histomorphometric, of pancreatic and renal tissues and the redox state in diabetic rats induced by STZ, supplemented or not with creatine. The literature shows several studies describing the presence of clinical and biochemical parameters [23,27,37,38,56,57,58]. However, creatine supplementation studies in DM without evaluating morphological parameters and tissue injury are quite recurrent. Therefore, the need for and importance of these analyses are evident.

During the saturation phase, five days before T1DM induction, creatine-supplemented animals from both groups received an average of 3 g/day (13–15 g/kg). Though low maintenance doses for more prolonged periods have the equivalent effect, this strategy is not possible due to the strong effects of STZ-induced DM1 in rats. Several studies use 2% of creatine diets to maintain its saturation levels on animal models [28], the same dose used in this study. For humans, recommended maintenance doses vary, ranging from 2–5 g/day [29,30]. Therefore, considering the difference in BMR [46] we believe that is pertinent to extrapolate that the used creatine dosage was adequate for evaluation as an adjuvant for DM in this experimental model.

Our results demonstrate that creatine supplementation attenuates several biochemical indicators in STZ-induced DM, such as blood glucose, serum urea, reduction in diabetes-induced ALT increase, and the restoration of the antioxidant enzymes activities and hydrogen peroxide content to control levels. The glycemia reduction found here and in other studies [23,37], and the maintenance of renal function per si strongly suggest that some diabetes therapies could benefit from this compound’s positive effects.

In addition, the histomorphometric analyses demonstrated that the creatine supplemented diabetic animals (DCr group) had an increased pancreatic islet area compared to the diabetic animals without the supplementation (D group). These results may be related to the fasting blood glucose levels in the DCr group being significantly lower than the D group and previous reports that creatine supplementation could reduce hyperglycemia in DM [23].

These findings may also be due to a direct antioxidant effect of this compound on ROS, exerting a protective or regenerative effect on the islets during DM or STZ treatment [28,59,60,61]. The restoration of the H_2_O_2_ content to control levels in the DCr animals reinforces this hypothesis. In addition, our creatine supplementation results on CAT, SOD, and GPx activities support the findings of Stefani et al. (2014) [62], which demonstrated that creatine was able to modulate antioxidant enzyme activity in rats undergoing resistance training.

The histopathological results here conform with the literature due to the identification of destroyed serous acini, ductal ectasia, and fibrosis in the diabetic groups [63,64,65,66,67], with the DCr group having the most extensive affected areas, which may be due to supplementation in animals already injured by STZ administration [68,69,70,71]. According to previous reports, inflammatory cell infiltrates, and necrosis observations are evident in DM, and we found both in the D and DCr groups. Creatine supplemented animals could not reduce the inflammatory parameter but could reduce necrosis, as observed for the DCr group.

However, it is possible to notice a possible detrimental effect of creatine while the CCr group also presented histopathological characteristics of tissue injury. In addition to this, the CCr group differed statistically from the other groups for the adipose replacement and hyperemia results. We could argue that creatine supplementation might have caused these effects, but on the other hand, previous studies have reported these findings as standard features in DM [72,73].

Based on these findings, we could diagnose the animals in the CCr, D, and DCr groups with pancreatitis, some with acute and others with chronic. Indeed, it is a frequent diagnosis, especially for diabetes, with most subclinical cases [74], as the signs are commonly subtle and nonspecific [66]. Therefore, we can affirm that creatine did not attenuate or reverse this specific condition in the DCr group. Instead, it seems to cause tissue alterations in the CCr group. Mohapatra et al. (2015) defined this T1DM-related subclinical fibrosis and exocrine dysfunction without symptoms of chronic pancreatitis as diabetic exocrine pancreatopathy, as seen in this present study’s results [64].

We also detected a significant reduction in the renal glomeruli counts in the DCr group when compared to the other experimental groups. This finding may be related to the histopathological observations of hyalinosis and glomerular ischemia that may be responsible for the destruction of this structure. The constitutive expression of the creatine transporter (CreaT) in kidneys [24,75,76] may favor this tissue damage. Moreover, the injuries and consequences of DM make the glomerular filtration barrier (GFB) more permeable, leading to an increase in protein filtration and fixation in the tubular lumen [77]. Because of this and the presence of CreaT in the kidneys, creatine or creatinine may have been hyper-filtered and condensed in the arterial and tubular lumen, leading to the impaired glomerular blood supply and ischemia. These results corroborate the higher urea concentrations in the D and DCr groups, the significant presence of cylindrical protein tubules in the CCr group, and the glomerular ischemia in the D, DCr, and CCr groups. Altogether, these changes support the influence of the pathological condition and indicate a probable effect of creatine supplementation in STZ-induced T1DM rats.

As a consequence of ischemia, the glomeruli lose podocytes, the epithelial cells of the filtration apparatus, leading to hyalinization of the glomerulus, which were only evident in the histological slides of the DCr group. This condition preferentially affects the renal tubules and leads to increased renal damage, since nephron loss by tubular necrosis may occur due to its obstruction by protein cylinders [77,78]. In addition, the decreased oxygen supply and intense arteriolar vasoconstriction may be an attempt to maintain normal blood pressure, as observed in the CCr, D, and DCr groups, which corroborate the findings of Obineche et al. (2001) [78].

It is also noteworthy the result of glomerular areas, which shows the DCr group similar to the control groups. On the other hand, group D was statistically different from the other groups, thus presenting smaller areas of the renal glomeruli. This result may reflect in the DCr group is a compensatory increase to mesangial expansion to preserve the filtration surface [79], while the diabetic non-supplemented group (D) presents glomerular atrophy due to hyalinization.

The thickening of the glomerular mesangial matrix consists of diffuse intercapillary glomerulosclerosis and precedes the nodular type, pathognomonic of diabetes. It is the strongest correlation with renal dysfunction in DM and is present in most patients with T1DM with diabetic nephropathy [79,80]. According to Najafian and Mauer (2012) [79], there may be mesangial expansion without basement membrane thickening, attesting to this present study’s histopathological findings, in which groups D and DCr showed only glomerulosclerosis.

However, the results of the glomerular area differ from those of Østerby and Gundersen (1975), Craven et al. (1997), and Salgado et al. (2004) [81,82,83]. These authors showed that the glomerulus increased both in the humans and animals models. Perhaps our findings may be due to the advanced nephron damages since glomerular hypertrophy commonly occurs in early DM stages [80]. According to the histopathological diagnosis, groups D and DCr present diffuse intercapillary glomerulosclerosis, tubular necrosis, glomerulonephritis, and diabetic nephropathy. Therefore, there is not an early disease stage.

The renal injuries found might induce immune cell deposition and complement fixation. According to the work of McPhee and Ganong (2011) [84], the prolonged inflammatory response may further destroy the glomerular architecture that, if not treated, can evoke irreversible end-stage renal damage marked by excessive immune cell invasion [77]. These authors also proposed that macrophage infiltration, stimulated by AGEs or tissue damage, can overexpress type IV collagen, causing structural changes through protein accumulation, resulting in glomerulosclerosis in models of T1DM. These observations support our findings and the higher intensity of inflammatory cells’ foci on histological slides in the D and DCr groups. However, histopathological changes can develop without clinical evidence [79]. Significant glomerular tissue lesions may be present while patients are normo- or microalbuminuric and with preserved glomerular filtration rate [85,86], which might explain the acceptable range for creatinine and serum urea concentrations in creatine-supplemented non-diabetic animals (CCr). Indeed, this group presented tubular necrosis but with no alteration on clinical markers.

Hyperglycemia, which entails increased water consumption, weight reduction, and polyphagia due to the inefficient use of glucose in tissues due to low insulin levels, is common in T1DM. Several studies had reported that creatine supplementation could reduce blood glucose [23,37,45,57,84,85,86,87,88]. Additional pieces of evidence point to an increased expression and translocation of GLUT-4 in skeletal striated muscle cells [87,89]. We did observe the reduced blood glucose in creatine-supplemented diabetic animals (DCr group). However, we also observed polydipsia, polyphagia, and decreased body weight in STZ-induced diabetic animals, similar to previous reports [90,91,92,93,94]. Therefore, creatine supplementation in diabetic animals neither attenuated nor aggravated these clinical effects. We emphasize this observation, especially for polydipsia, since creatine is an osmotically active substance [95]. We hypothesized that the higher intramuscular water retention and slower release to the extracellular medium would lead to dehydration [96] and interfere with this outcome. Nevertheless, as observed elsewhere [30,97,98], the water balance was not affected.

Even in creatinine serum concentration, a widely used clinical marker to assess kidney function [35,53], and altered during non-treated DM [99,100], we did not find any significant difference between the experimental groups. These findings, along with the reduction in serum urea concentration in animals from the DCr group, demonstrate that creatine does attenuate kidney function in the supplemented animals, reinforcing the results obtained by de Souza and Silva (2019) and Persky and Rawson (2007) [27,101]. However, the morphological analyses also demonstrate that creatine does not alleviate histopathological damages to a similar extent, especially for renal tissues. It is also noteworthy that these alterations are probably transitory and did not compromise the organ’s function as a whole.

Similar to urea, transaminases, especially ALT and AST, are enzymes associated at high levels with markers of liver injury or necrosis [102,103]. AST showed no significant difference between the groups. On the other hand, diabetic groups presented increased ALT concentration, indicating tissue injuries by the DM itself or by STZ administration [68,104]. It is of particular interest that ALT is essential in the liver and kidney gluconeogenesis, transforming alanine into pyruvate for glucose production [105,106,107], being critical for DM. Therefore, our results could indicate that the significant ALT reduction in the animals of the DCr group, compared to the D group, might have a protective effect on the liver and kidney tissues.

The low level of oxidative damage in proteins in the D group and the no change of this parameter in the DCr group can be explained by the kidney’s high rate of protein renewal. Besides, diabetes itself induces molecular remodeling through proteases, which could modulate the levels of this marker [108]. Portero-Otín (1999) [109] suggested that the decrease or no change in carbonylated proteins in diabetic rats may be due to increased protease activity in oxidatively altered proteins. Furthermore, in general, the degradation of oxidatively modified proteins is prioritized over native proteins [110]—a fact that reinforces our cited hypotheses.

This work has some limitations. First, despite its widespread use for T1DM investigations, a single STZ injection may affect the pancreatic β-cells heterogeneously, mainly due to β-cell glucotoxicity. Thus, diabetic animals may exhibit distinct levels of hyperglycemia or even some degree of islets regeneration [111]. Another potential limitation of our study is that we performed analyses at a single point, and asynchronous responses between biochemical and histopathological responses may have occurred throughout the experimental period. However, other reports demonstrate that a 30-day trial is sufficient to produce several systemic damages from the chronic hyperglycemia that sets in [112,113]. In addition, we used a single-one supplementation scheme, and there is no agreement on the creatine dosage for rats in the literature. Even though we are optimistic that the creatine doses used in this study are adequate to the model, we cannot exclude that this dosage is high enough to exert renal and pancreatic tissues damages.

## 5. Conclusions

According to the biochemical and morphological parameters found in this study, it is evident that creatine supplementation could significantly improve the consequences of STZ-induced DM, as observed by the attenuation in biochemical parameters. Despite this improvement, the morphological analyses of pancreatic and renal tissues indicated that the supplementation did not attenuate at the same extent the histopathological alterations from the disease. Therefore, we emphasize the importance of future studies to better understand the effects of creatine supplementation in the pancreatic and renal diabetic tissues and on the biochemical parameters in the long term.

## Figures and Tables

**Figure 1 nutrients-14-00431-f001:**
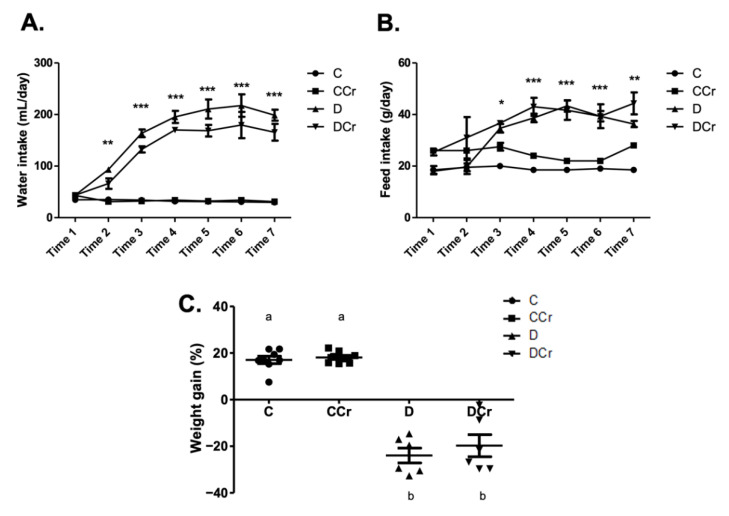
Classic symptoms of DM1. Daily measurement of (**A**) water intake and (**B**) feed consumption of groups C, CCr, D, and DCr groups, at seven evaluation points (weeks) during the 40-day experimental protocol. (**C**) Quantification of weight gain (C = 8; CCr = 8; D = 6; DCr = 6) at the end of the 40-day experimental period. Graph shows mean ± SEM. * *p* < 0.05 for C and CCr vs. D and DCr. ** *p* < 0.01 for C and CCr vs. D and DCr. *** *p* < 0.001 for C and CCr vs. D and DCr. C) Weight gain: *p* < 0.001. C—normoglycemic non-supplemented animals; CCr—normoglycemic creatine supplemented animals; D—non-supplemented STZ-induced diabetics animals; DCr—creatine-supplemented STZ-induced diabetics animals. Time = week.

**Figure 2 nutrients-14-00431-f002:**
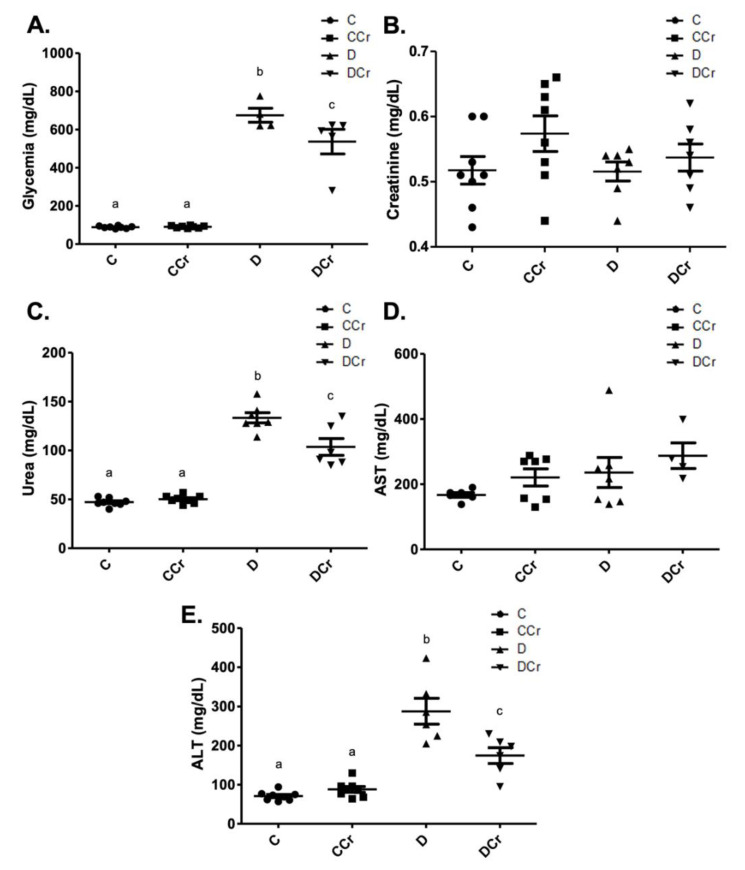
Biochemical parameters evaluated—(**A**) blood glucose (C = 8; CCr = 8; D = 5; DCr = 5), (**B**) serum creatinine, (**C**) serum urea (C = 8; CCr = 8; D = 7; DCr = 6), (**D**) aspartate aminotransferase (AST) (C = 6; CCr = 7; D = 7; DCr = 4), and (**E**) serum alanine aminotransferase (ALT) (C = 8; CCr = 8; D = 6; DCr = 6) of the experimental groups at the end of the experimental protocol. The graph represents the mean ± SEM. The divergent letters represent mean significant differences compared to the control group. (**A**) b *p* < 0.001 and c *p* < 0.05. (**C**) b *p* < 0.001 and c *p* < 0.001. (**E**) b *p* < 0.001 and c *p* < 0.01. C—normoglycemic non-supplemented animals; CCr—normoglycemic creatine supplemented animals; D—non-supplemented STZ-induced diabetics animals; DCr—creatine-supplemented STZ-induced diabetics animals.

**Figure 3 nutrients-14-00431-f003:**
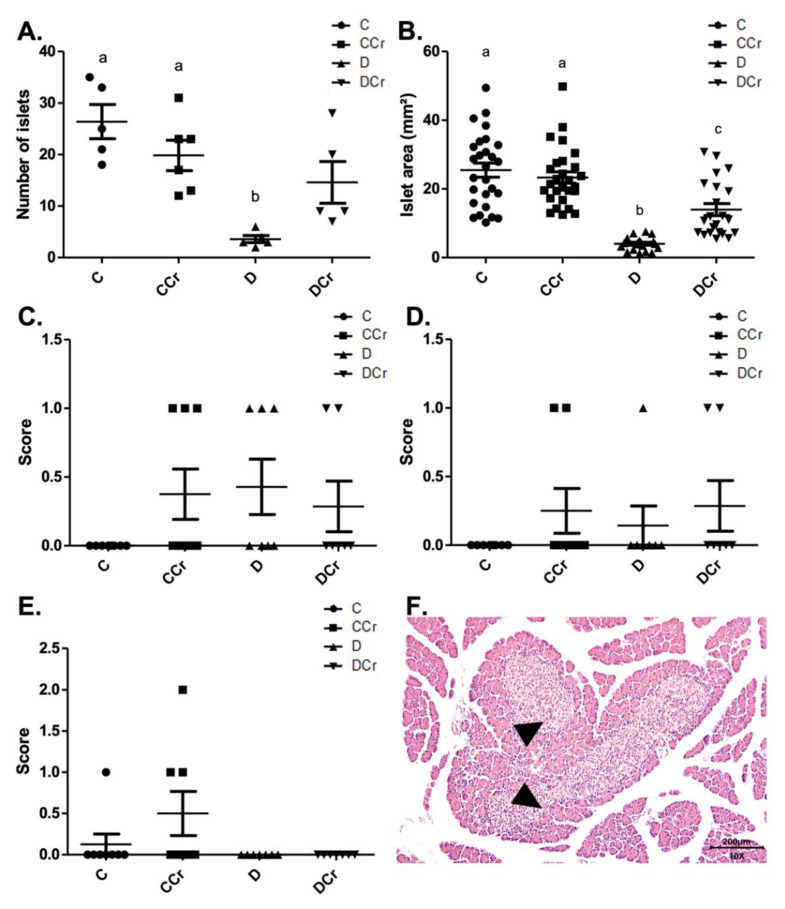
Histomorphometry of the pancreas for all experimental groups. (**A**) The number of pancreatic islets C (*n* = 5); CCr (*n* = 6); D (*n* = 5); DCr (*n* = 5); (**B**) Area (mm^2^) of pancreatic islets; (**C**) parenchymal atrophy; (**D**) pancreatic duct alteration; (**E**) Pancreatic islet hyperplasia and (**F**) Hyperplastic pancreatic islet (triangles) in the 10× objective, at 200 μm scale, were observed for all experimental groups. The graph shows mean ± SEM. The divergent letters represent mean significant differences. (**A**) b *p* < 0.001. (**B**) b *p* < 0.001; c *p* < 0.01. C—normoglycemic non-supplemented animals; CCr—normoglycemic creatine supplemented animals; D—non-supplemented STZ-induced diabetics animals; DCr—creatine-supplemented STZ-induced diabetics animals.

**Figure 4 nutrients-14-00431-f004:**
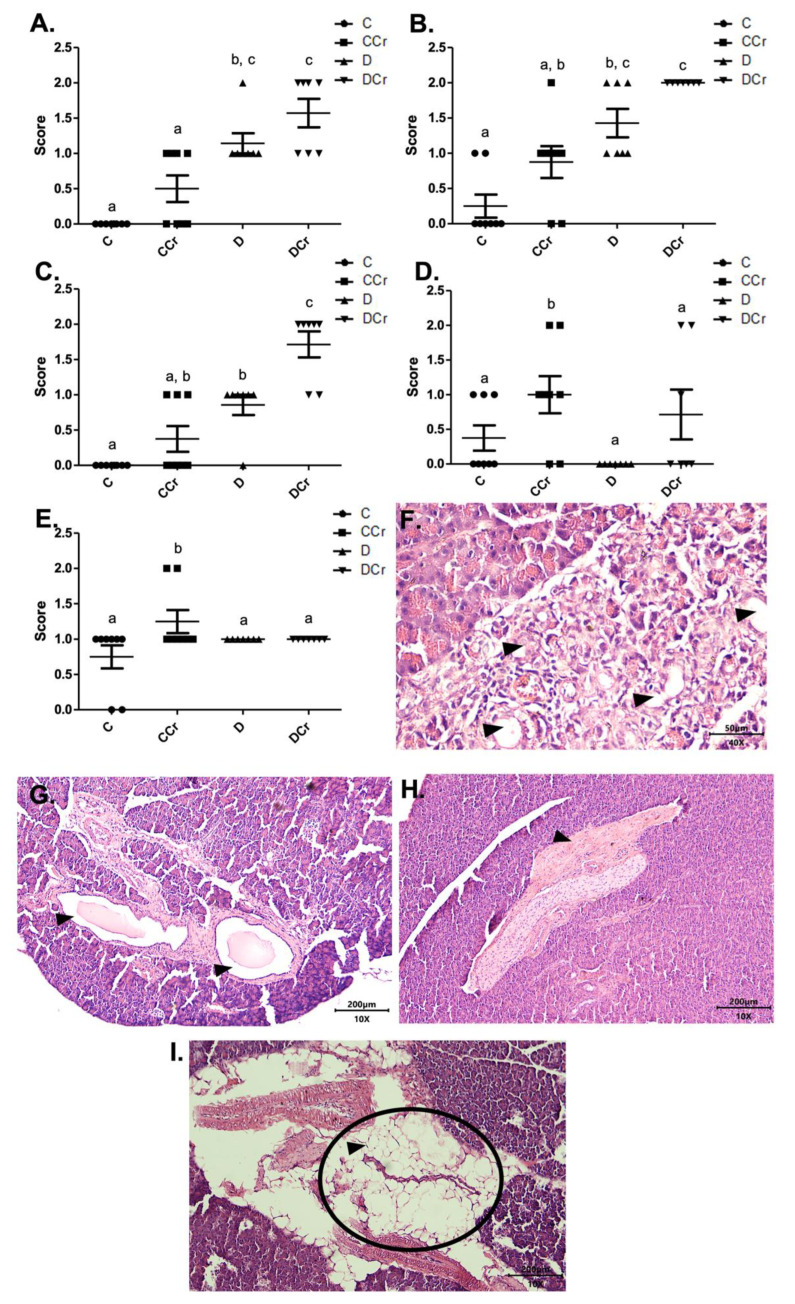
Identification of (**A**) destruction of pancreatic acini, (**B**) pancreatic ductal ectasia, (**C**) presence of pancreatic fibrosis, (**D**) adipose replacement in the pancreas and (**E**) hyperemia in the pancreatic tissue for the experimental groups.The graph shows mean ± SEM. The divergent letters represent mean significant differences compared to the control group (**A**) b *p* < 0.05; c *p* < 0.001; (**B**) b *p* < 0.05; c *p* < 0.001; (**C**) b *p* < 0.01; c *p* < 0.001; (**D**,**E**) b *p* < 0.05. (**F**) Photomicrograph of destroyed pancreatic acini. Slide stained with HE, at 40× objective and represented on a 50 μm scale. The tips show white spaces due to the destruction of the acinar cells. (**G**) Photomicrograph of ductal ectasia. HE-stained slide, 10× objective and represented on a 200 μm scale. The triangles point to ductal ectasia (dilatation of pancreatic ducts). (**H**) Photomicrograph of pancreatic fibrosis. HE stained the slide at 10× objective and represented on a 200 μm scale. At the tip of the triangle, there is a fibrotic area in the pancreatic parenchyma. (**I**) Photomicrograph showing adipose replacement in the pancreatic tissue. Blade stained in HE at 10× magnification and represented on a 200 μm scale. Tip of the triangle: adipocyte. Ellipse: a cluster of adipocytes in the pancreatic parenchyma. C—normoglycemic non-supplemented animals; CCr—normoglycemic creatine supplemented animals; D—non-supplemented STZ-induced diabetics animals; DCr—creatine-supplemented STZ-induced diabetics animals.

**Figure 5 nutrients-14-00431-f005:**
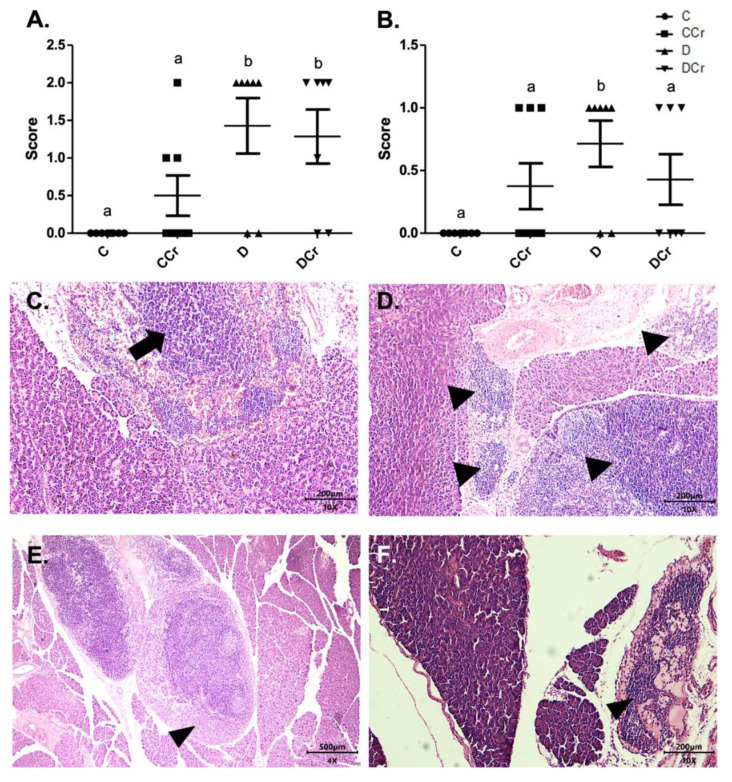
(**A**) Mononuclear inflammatory infiltrate present in the pancreatic tissue and (**B**) Presence of necrosis in the pancreatic parenchyma in the experimental groups. The graph shows mean ± SEM. The divergent letters represent mean significant differences compared to the control group (**A**) b *p* < 0.01; (**B**) b *p* < 0.05. (**C**) Photomicrograph of intense inflammatory cell infiltrates in the exocrine part of the pancreas. Blade stained in HE, on 10× objective and represented on a 200 μm scale. Arrow: points to the intense infiltrate of inflammatory cells. (**D**) Photomicrograph of foci of inflammatory infiltrates in the exocrine portion of the pancreas. The tips of the triangles show the foci of inflammatory cells. Slide stained with HE, 10× objective of 10× and represented in a 200 μm scale. (**E**) Photomicrograph of extensive necrosis in the pancreatic parenchyma. Slide stained in HE, with a 4× objective 4× objective and represented on a 500 μm scale. (**F**) Photomicrograph of necrosis in the pancreatic parenchyma stained in HE, in the 10× objective and 200 μm scale. The tips of the triangles show necrosis in the parenchyma. C—normoglycemic non-supplemented animals; CCr—normoglycemic creatine supplemented animals; D—non-supplemented STZ-induced diabetics animals; DCr—creatine-supplemented STZ-induced diabetics animals.

**Figure 6 nutrients-14-00431-f006:**
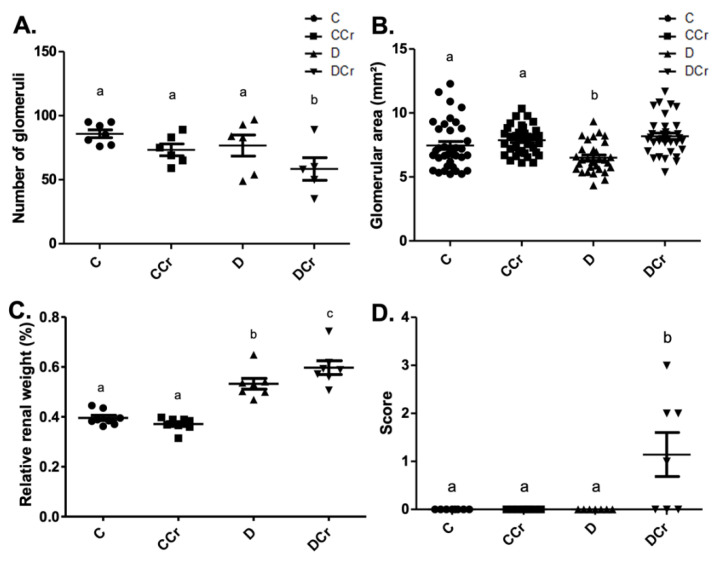
Histomorphometry of renal tissue regarding (**A**) Number of renal glomeruli; C (*n* = 7); CCr (*n* = 6); D (*n* = 6); DCr (*n* = 5). (**B**) Area of renal glomeruli, in mm^2^, (**C**) Relative renal weight at the end of the 40-day experimental period of the experimental groups. (**D**) Presence of glomerular hyalinosis. The graph presents mean ± SEM. The divergent letters represent mean significant differences compared to the control group. (**A**) b *p* < 0.05; (**B**) b *p* < 0.05; (**C**) b *p* < 0.001 and c *p* < 0.001; (**D**) b *p* < 0.01. C—normoglycemic non-supplemented animals; CCr—normoglycemic creatine supplemented animals; D—non-supplemented STZ-induced diabetics animals; DCr—creatine-supplemented STZ-induced diabetics animals.

**Figure 7 nutrients-14-00431-f007:**
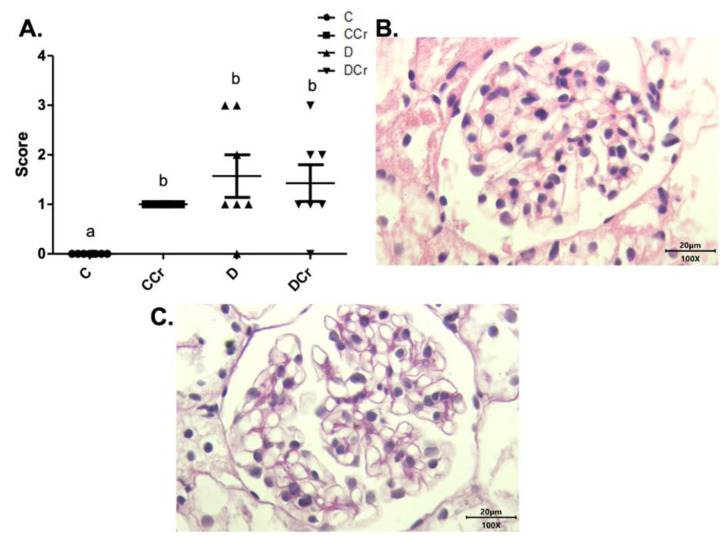
(**A**) Ischemia in renal glomeruli in the experimental groups. The graph shows mean ± SEM. The divergent letters represent mean significant differences compared to the control group. b *p* < 0.05. (**B**) Photomicrograph of the renal glomerulus with ischemia. Stained in HE, 100× objective and represented on a 20 μm scale. (**C**) Photomicrograph of the ischemic renal glomerulus. Stained in PAS, 100× objective and represented on a 20 μm scale. C—normoglycemic non-supplemented animals; CCr—normoglycemic creatine supplemented animals; D—non-supplemented STZ-induced diabetics animals; DCr—creatine-supplemented STZ-induced diabetics animals.

**Figure 8 nutrients-14-00431-f008:**
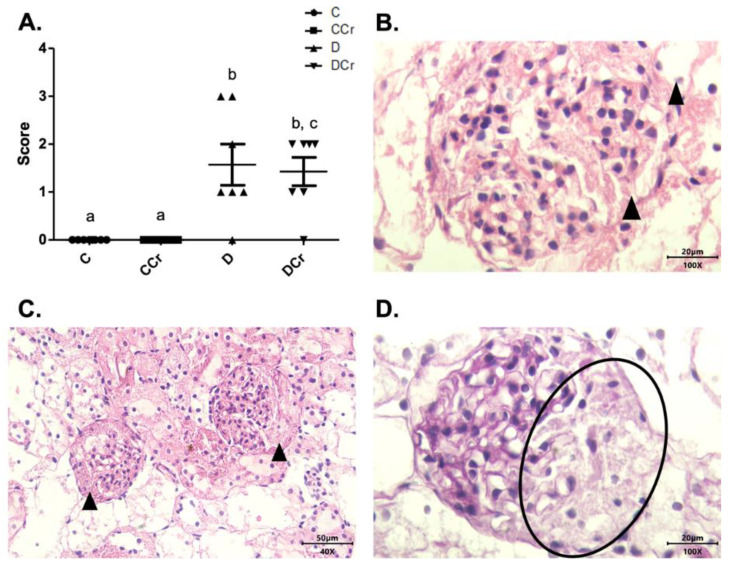
(**A**) Histopathological identification of diffuse intercapillary glomerulosclerosis in the experimental groups. The graph shows mean ± SEM. The divergent letters represent mean significant differences compared to the control groups. b *p* < 0.001; c *p* < 0.01. (**B**) Histopathology photomicrograph. HE stained the slide on 100× objective and a 20μm scale. The tip of the triangle shows diffuse thickening of the mesangial matrix. (**C**) Photomicrograph of diffuse intercapillary glomerulosclerosis. HE-stained slide, in 40× objective and on a 50 μm scale. At the tip of the triangle: Diffuse thickening of the mesangial matrix. (**D**) Photomicrograph of diffuse intercapillary glomerulosclerosis. Slide PAS stained, at 40× magnification and on a 20 μm scale. Ellipse: highlights the thickening of the mesangial matrix. C—normoglycemic non-supplemented animals; CCr—normoglycemic creatine supplemented animals; D—non-supplemented STZ-induced diabetics animals; DCr—creatine-supplemented STZ-induced diabetics animals.

**Figure 9 nutrients-14-00431-f009:**
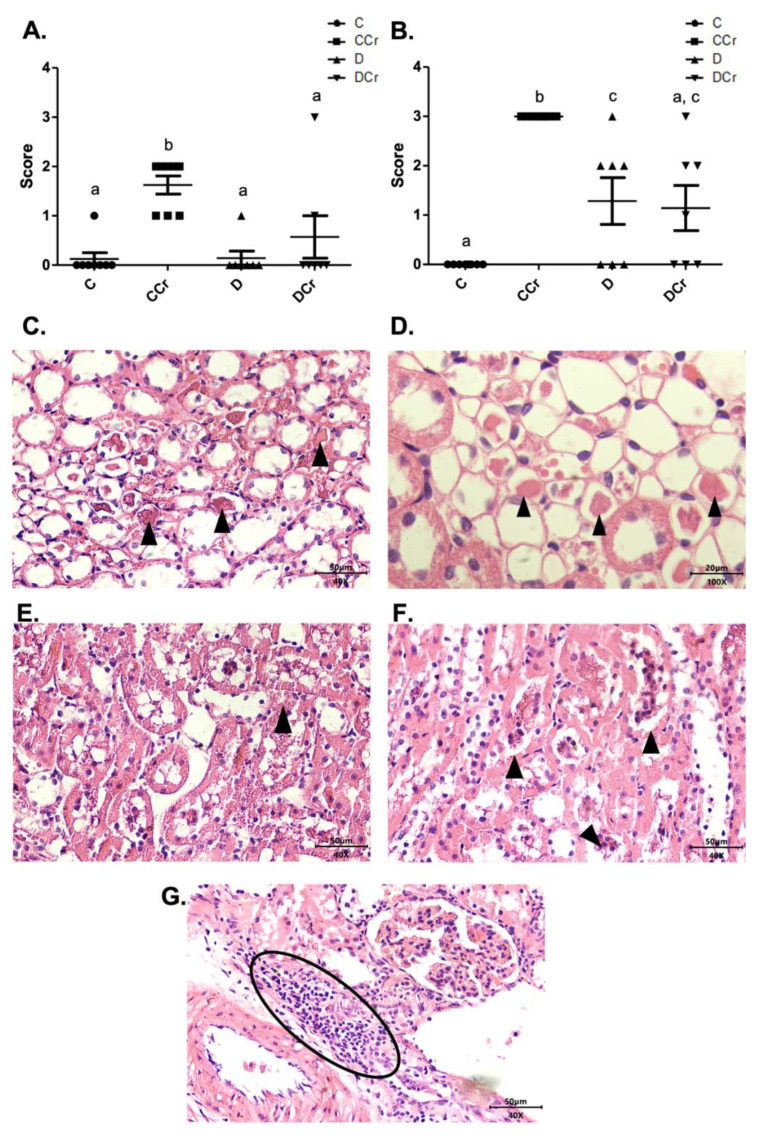
(**A**) Protein cylindrical tubules or hyaline cylinders and (**B**) necrotic tubules are present in renal tissue for all experimental groups. The graph presents mean ± SEM. The divergent letters represent mean significant differences compared to the control group. (**A**) b *p* < 0,01; (**B**) b *p* < 0.01; c *p* < 0.05. (**C**) Photomicrograph of renal tubules with the presence of proteins in their lumen. Slide stained in HE, 40× objective 40× objective and on a 50 μm scale. (**D**) Photomicrograph of the renal tubules with the presence of proteins in their light. Slide stained in HE, objective 100×, scale 20 μm. The tips of the triangles show the presence of protein/hyaline cylinders in the renal tubules. (**E**) Photomicrograph of tubular necrosis in renal tissue. HE stained the slide at 40× magnification and on a 50 μm scale. (**F**) Photomicrograph of the presence of tubular necrosis in renal tissue. Slide stained at HE, 40× magnification, 50 μm scale. The tips of the triangles show necrotic cells present in the tissue. (**G**) Photomicrograph of the presence of inflammatory infiltrates in renal tissue. Slide stained in HE, 40× objective and on a 50 μm scale. The ellipse highlights the inflammatory cells present in the tissue. C—normoglycemic non-supplemented animals; CCr—normoglycemic creatine supplemented animals; D—non-supplemented STZ-induced diabetics animals; DCr—creatine-supplemented STZ-induced diabetics animals.

**Figure 10 nutrients-14-00431-f010:**
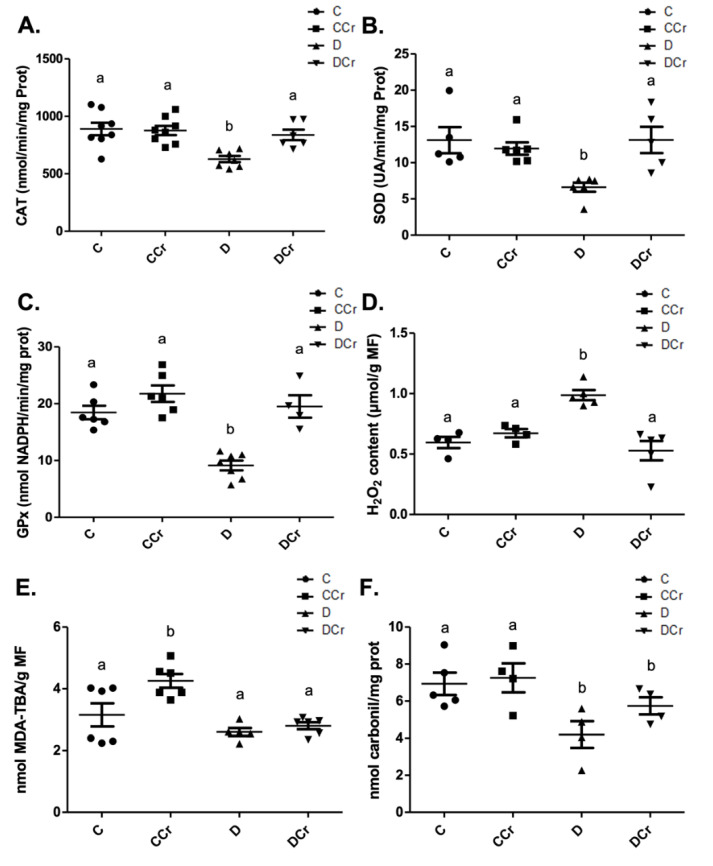
(**A**) Catalase (C = 8; CCr = 8; D = 7; DCr = 6), (**B**) superoxide dismutase (C = 5; CCr = 6; D = 6; DCr = 5) and (**C**) glutathione peroxidase renal activity (C = 6; CCr = 6; D = 7; DCr = 4). (**D**) Hydrogen peroxide content (C = 4; CCr = 4; D = 5; DCr = 5), (**E**) thiobarbituric acid reactive substances (C = 6; CCr = 6; D = 5; DCr = 6), (**F**) carbonylated protein content (C = 5; CCr = 4; D = 4; DCr = 4) renal of the experimental groups at the end of the experimental protocol. The graph represents mean ± SEM. The divergent letters represent mean significant differences compared to the control group. (**A**) b *p* < 0.01; (**B**) b *p* < 0.01; (**C**) b *p* < 0.001; (**D**) b *p* < 0.01. (**E**) b *p* < 0.001. (**F**) b *p* < 0.05. C—normoglycemic non-supplemented animals; CCr—normoglycemic creatine supplemented animals; D—non-supplemented STZ-induced diabetics animals; DCr—creatine-supplemented STZ-induced diabetics animals.

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
