# Peer review of "Effects of Creatine Supplementation on Histopathological and Biochemical Parameters in the Kidney and Pancreas of Streptozotocin-Induced Diabetic Rats"

_nutrients, 2022, doi:10.3390/nu14030431_

Round 1
Reviewer 1 Report
I am a bit puzzled by this paper. On the one hand, the authors report both favourable and unfavourable effects of creatine, but their comments are unexplainably tilted towards emphasizing negative results only. Furthermore, several issues should be fixed concerning data presentation and description. Specific comments follow.
- In creatine-supplemented diabetic rats the authors reported decreased blood glucose (line 237), decreased serum urea (line 241), reduction of diabetes-induced ALT increase (line 245), increase in number of pancreatic islets (fig. 3A), increase in volume of pancreatic islets (fig. 3B), prevention of diabetes-induced reduction in antioxidant enzymes (lines 438-439). All of these represent significant and beneficial effects of creatine supplementation, suffice it to say that glycaemia control is the main target of all diabetes therapies. Yet, the authors are unexplainably biased towards emphasizing harmful, histopathological findings. A much more balanced discussion is mandatory.
- The above observation is even more cogent since the authors report no difference in creatinine level between creatine-supplemented and creatine-not-supplemented animals (line 238). This fact, coupled to the decreased blood urea concentration in creatine-supplemented animals, point to a conservation of kidney function. This questions the real clinical relevance of the morphological damages that the authors observed: if necrosis and degeneration of kidney tissue does not end up into functional (creatinine and urea) damage, what is the real clinical relevance of the morphological damages they found? Perhaps not much.
- The authors should convey an idea of how much kidney parenchyma was affected by the negative changes they describe. This is especially necessary in view of the above comment.
- 6 cannot be understood, because the authors do not explain what is statistically different from what. They report various degrees of significance, but they do not explain to what comparisons they refer.
- Groups D and DCr in fig. 7B have no error bar. Does this mean that they each have N=1? If so, the whole figure is rather meaningless…
- Still concerning fig. 7, the authors do not report what “a” means. Furthermore, they do not report to what comparisons “a” and “b” refer.
- Lines 238-239, RE: “The significant difference seen in group D (p<0.05) was probably due to the higher number of slides with this finding”. It is unclear what this statement means.
- 11 and 12 report changes observed only in creatine-supplemented, diabetic rats. This is weird, and the authors should comment on that. Lacking a credible explanation, these figures should be deleted.
- 13, fig. 14 and fig. 15: definition of “a” is missing. Moreover, here, too, the authors do not explain to what comparisons “a” and “b” refer to.
- I recommend that the authors review all their figure legends, to make sure that the necessary statistical informations are included.
- In the conclusions, the sentence “According to all biochemical and morphological parameters found in this study, it is evident that creatine supplementation could not significantly improve the consequences of STZ-induced DM” is obviously wrong, because of the many positive effects of creatine (see above) that the authors report.
- Lines 220-221: there is conflict between “there was no significant difference” and “p<0.01”. One of the two statements must be a lapsus.
Author Response
Comments and Suggestions for Authors are in bold/italics. Our responses are in blue.
"I am a bit puzzled by this paper. On the one hand, the authors report both favourable and unfavourable effects of creatine, but their comments are unexplainably tilted towards emphasizing negative results only. Furthermore, several issues should be fixed concerning data presentation and description. Specific comments follow."
The authors would like to thank the reviewer for the thorough revision of our manuscript. His report made us rethink a more balanced way to describe the effects of creatine supplementation, as we will explain in context to each observation.
"In creatine-supplemented diabetic rats the authors reported decreased blood glucose (line 237), decreased serum urea (line 241), reduction of diabetes-induced ALT increase (line 245), increase in number of pancreatic islets (fig. 3A), increase in volume of pancreatic islets (fig. 3B), prevention of diabetes-induced reduction in antioxidant enzymes (lines 438-439). All of these represent significant and beneficial effects of creatine supplementation, suffice it to say that glycaemia control is the main target of all diabetes therapies. Yet, the authors are unexplainably biased towards emphasizing harmful, histopathological findings. A much more balanced discussion is mandatory.
The above observation is even more cogent since the authors report no difference in creatinine level between creatine-supplemented and creatine-not-supplemented animals (line 238). This fact, coupled to the decreased blood urea concentration in creatine-supplemented animals, point to a conservation of kidney function. This questions the real clinical relevance of the morphological damages that the authors observed: if necrosis and degeneration of kidney tissue does not end up into functional (creatinine and urea) damage, what is the real clinical relevance of the morphological damages they found? Perhaps not much."
Thank you for bringing our attention to this point. Even though the authors are from distinct disciplinary fields, the text highlights the morphological damages. However, we still consider our morphological and histopathological findings clinically relevant. Glycaemia, transaminases, serum creatinine, and urea levels are important biochemical parameters, indicating improvement of diabetes in creatine supplemented animals. Nevertheless, they can also be misleading under specific complex metabolic contexts, such as the one used in our experimental design. In addition, these parameters are an integrative result of the alterations in other organs, such as the liver and skeletal muscle, whose responses we did not evaluate.
There is indeed an alleviation of the kidney function in creatine-supplemented diabetic rats. Still, the induced morphological consequences (predominantly renal glomeruli ischemia and diffuse intercapillary glomerulosclerosis) could lead to other severe problems, especially a more accentuated loss of kidney function under more prolonged supplementation treatments. Even with the aid in blood glucose control, these creatine-induced observations could sum up the already known DM consequences to renal tissues. Since we only took measurements at just one time, we can not predict the further progress of these morphological alterations or if they can later lead to impacts on the aforementioned biochemical parameters. Therefore, we decided to be careful, as pointed by the reviewer. We intended to give a detailed view of the evidenced changes, though they are probably transitory, and, at the time observed, they did not compromise the organ's function as a whole. Indeed, this unbalanced discussion could mislead the readers that we overlooked the potential of creatine supplementation as an adjuvant to DM treatment. Instead, we believe that its supplementation could effectively help specific DM cases.
"The authors should convey an idea of how much kidney parenchyma was affected by the negative changes they describe. This is especially necessary in view of the above comment."
We thank the reviewer for this well-timed consideration. An idea of how much of affected parenchyma will shed light on the above discussion. However, we consider that it could be a subjective observation. So, we opted to replace the bar plots with point graphs, giving a more precise appreciation of the morphological and histopathological observations, as well as the sample size.
"6 cannot be understood, because the authors do not explain what is statistically different from what. They report various degrees of significance, but they do not explain to what comparisons they refer.
Groups D and DCr in fig. 7B have no error bar. Does this mean that they each have N=1? If so, the whole figure is rather meaningless…
Still concerning fig. 7, the authors do not report what “a” means. Furthermore, they do not report to what comparisons “a” and “b” refer."
We agree with the reviewer, and we made the necessary changes in figures and legends. We also re-organized the figures, as pointed out above.
"Lines 238-239, RE: “The significant difference seen in group D (p<0.05) was probably due to the higher number of slides with this finding”. It is unclear what this statement means."
We fully agree with this observation. We removed the sentence.
"11 and 12 report changes observed only in creatine-supplemented, diabetic rats. This is weird, and the authors should comment on that. Lacking a credible explanation, these figures should be deleted.
13, fig. 14 and fig. 15: definition of “a” is missing. Moreover, here, too, the authors do not explain to what comparisons “a” and “b” refer to.
I recommend that the authors review all their figure legends, to make sure that the necessary statistical information are included."
We gratefully appreciate the recommendations. We joined figures 10 and 12 and removed the plot from figure 11 (since it was not significant). We also joined the figures 15, 16, and 17, removing the A plot from this latter one. We revised and improved all the figures' legends. Please, find below the list of all modifications:
- We joined figures 3, 4, and 5 in the new Figure 3. We reduced the number of plots containing non-significant results.
- Figures 6 and 7 and 8 and 9 are now the new Figure 4 and Figure 5, respectively.
- We joined the figures 10, 11, and 12. Now, they are the new Figure 6. We excluded Figure 11.
- Figures 13 and 14 were maintained as figures 7 and 8, respectively.
- The new Figure 9 resulted from figures 15 and 16. We removed figure 17.
- We merge all the oxidative status (Figures 18 and 19) results in Figure 10.
"In the conclusions, the sentence “According to all biochemical and morphological parameters found in this study, it is evident that creatine supplementation could not significantly improve the consequences of STZ-induced DM” is obviously wrong, because of the many positive effects of creatine (see above) that the authors report."
We rewrote the conclusion in the shed of all the reviewer's considerations and keen observations. The rewrote conclusion:
“According to the biochemical and morphological parameters found in this study, it is evident that creatine supplementation could significantly improve the consequences of STZ-induced DM, as observed by the attenuation in biochemical parameters. Despite this improvement, the morphological analyses of pancreatic and renal tissues indicated that the supplementation did not attenuate at the same extent the histopathological alterations from the disease. Therefore, we emphasize the importance of future studies to better understand the effects of creatine supplementation in the pancreatic and renal diabetic tissues and on the biochemical parameters in the long term.”
" Lines 220-221: there is conflict between “there was no significant difference” and “p<0.01”. One of the two statements must be a lapsus."
Thanks for noticing this incoherence. It was indeed a typo.
Reviewer 2 Report
The work from Goncalves and co-workers described the effect of Creatine supplementation in STZ treated rats. They examined in depth histological and biochemical changes in pancreatic and renal function. They concluded that creatine supplementation does not singnificantly improve STZ-induced effect at least in the two examinated organs. Moreover, they displayed creatine-dependent detrimental effect on histological parameters in normal pancreatic and renal tissue.
The study is well organized and the analyses are well performed. The manuscript could be improved according the following suggestions:
1) All bar graphs should be replaced with point graphs to appreciate sample size and creatine effects. Sample size should be indicated in each figure legend.
2) Too many figures are present in the present version of the manuscript. Figures should be re-organized and put together in multi-panel figures (i.e. Figs 3 - 4 - 5). 10 max 12 figures should be enough. At moment they are too fragmented.
3) A separate paragraph with study limitations should be added, for example authors used only a specific pharmacological DM-indiced model and performed their analysis only at a single time point.
4) Organs object of the present manuscript should be mentioned in the article title.
Author Response
Comments and Suggestions for Authors are in bold/italics. Our responses are in blue.
"The work from Goncalves and co-workers described the effect of Creatine supplementation in STZ treated rats. They examined in depth histological and biochemical changes in pancreatic and renal function. They concluded that creatine supplementation does not singnificantly improve STZ-induced effect at least in the two examinated organs. Moreover, they displayed creatine-dependent detrimental effect on histological parameters in normal pancreatic and renal tissue.
The study is well organized and the analyses are well performed."
The authors would like to thank the reviewer for the thorough revision of our manuscript. We appreciated his objective and keen observations, which will undoubtedly improve our manuscript.
"The manuscript could be improved according the following suggestions:
1) All bar graphs should be replaced with point graphs to appreciate sample size and creatine effects. Sample size should be indicated in each figure legend."
We accept the reviewer's suggestion. We replace all the bar plots to point graphs with mean±SEM representation. We also described the sample size for each group in each figure legend.
"2) Too many figures are present in the present version of the manuscript. Figures should be re-organized and put together in multi-panel figures (i.e. Figs 3 - 4 - 5). 10 max 12 figures should be enough. At moment they are too fragmented."
Another appreciated suggestion. We joined several figures together, and the manuscript now has ten figures. Please, find below the list of all modifications:
- We joined figures 3, 4, and 5 in the new Figure 3. We reduced the number of plots containing non-significant results.
- Figures 6 and 7 and 8 and 9 are now the new Figure 4 and Figure 5, respectively.
- We joined the figures 10, 11, and 12. Now, they are the new Figure 6. We excluded Figure 11.
- Figures 13 and 14 were maintained as figures 7 and 8, respectively.
- The new Figure 9 resulted from figures 15 and 16. We removed figure 17.
- We merge all the oxidative status (Figures 18 and 19) results in Figure 10.
"3) A separate paragraph with study limitations should be added, for example authors used only a specific pharmacological DM-indiced model and performed their analysis only at a single time point."
The reviewer is correct. We added a new paragraph with further elaboration and a critical view of our study's limitations. We also justified the single time point experimental design.
“This work has some limitations. First, despite its widespread use for type 1 DM investigations, a single one STZ injection may affect heterogeneously the pancreatic β-cells, mainly due to β-cell glucotoxicity. Thus, diabetic animals may exhibit distinct levels of hyperglycemia or even some degree of islets regeneration [1]. Another potential limitation of our study is that we performed analyses at a single time-point, and an imbalance between biochemical and histopathological alterations may have occurred throughout the experimental period. However, it is well known that a 30-days trial is sufficient to produce several systemic damages from the chronic hyperglycemia that sets in [2,3]”.
- Wu J, Yan LJ. Streptozotocin-induced type 1 diabetes in rodents as a model for studying mitochondrial mechanisms of diabetic β cell glucotoxicity. Diabetes Metab Syndr Obes. 2015 Apr 2;8:181-8.
- Peixoto JVC, Santos ASR Jr, Corso CR, da Silva FS, Capote A, Ribeiro CD, Abreu BJDGA, Acco A, Fogaça. Thirty-day experimental diabetes impairs contractility and increases fatigue resistance in rat diaphragm muscle associated with increased anti-oxidative activity. Can J Physiol Pharmacol. 2020 Aug;98(8):490-497.
- Castro VMD, Medeiros KCP, Lemos LIC, Pedrosa LFC, Ladd FVL, Carvalho TG, Araújo Júnior RF, Abreu BJ, Farias NBDS. S-methyl cysteine sulfoxide ameliorates duodenal morphological alterations in streptozotocin-induced diabetic rats. Tissue Cell. 2021 Apr;69:101483.
"4) Organs object of the present manuscript should be mentioned in the article title."
We have now rephrased the article title according to the reviewer’s recommendation:
“Effects of Creatine Supplementation on Histopathological and Biochemical Parameters in the Kidney and Pancreas of Streptozotocin-Induced Diabetic Rats”
Round 2
Reviewer 1 Report
The authors significantly improved their paper, by addressing the concerns I had expressed. However, a couple of points still need improvement:
1) in almost all figure legends they should clearly state what is significantly different from what. Although this can to an extent be understood from the text, it must be clearly said in the figure legends.
2) although the conclusions are now satisfactory, the discussion still does not mention the beneficial biochemical changes obtained in the creatine group. The authors should add a part (I suggest a subsection) even briefly discussing them
Author Response
The authors would like to thank the reviewer for improving our manuscript. Please see below the answer to the specific comments:
1) The significant differences are in comparison to the control group values. We added this information to the figure legends.
2) We added a new paragraph to the discussion, briefly highlighting the beneficial effects of creatine on the biochemical parameters. We did not opt for a specific subsection because some parts of the discussion integrate the histopathological findings to the biochemical parameters. We believe this integration is unexplored in the literature and necessary to understand creatine supplementation in diabetic and non-diabetic states.